# Flourishing among Children and Adolescents with Chronic Pain and Emotional, Developmental, or Behavioral Comorbidities

**DOI:** 10.3390/children10091531

**Published:** 2023-09-09

**Authors:** Madeline Foster, Jessica Emick, Nathan M. Griffith

**Affiliations:** School of Psychology, Fielding Graduate University, Santa Barbara, CA 93105, USA

**Keywords:** chronic pain, mental health, comorbidities, flourishing

## Abstract

Pediatric chronic pain is an important public health issue given its notable impact on numerous domains of living. Pediatric chronic pain is also often comorbid with emotional, developmental, or behavioral conditions, which can lead to more severe negative outcomes and an even greater reduction in positive outcomes compared to those without comorbidities. Flourishing is a positive outcome that chronic pain status has been shown to impact. Flourishing in children aged 6–17 years living with chronic pain, as well as those with chronic pain and comorbidities, was explored using data from the 2018/2019 National Survey of Child Health. Chronic pain occurred in 4.0% of our sample, and the prevalence of chronic pain plus comorbidities was 3.9%. There were significant associations between the chronic pain condition status and all demographic variables (sex, age, race/ethnicity, poverty level, parental education, and health insurance status). The results of the hierarchical logistic regression found that the chronic pain condition status significantly predicted flourishing. Children with chronic pain were 2.33 times less likely to flourish, and children with chronic pain plus an emotional, developmental, or behavioral comorbidity were 13 times less likely to flourish than their typical peers. Given their significantly lower likelihood of flourishing, there is an urgent need for interventions targeted at children experiencing chronic pain and mental health comorbidities.

## 1. Introduction

In their 2020 Guidelines on the Management of Chronic Pain in Children, the World Health Organization (WHO) identified chronic pain—pain that persists or recurs for more than three months—as the leading cause of morbidity in children [1]. The impact on a child living with chronic pain can be severe, deleterious, and far-reaching, with the literature finding school functioning, peer and social interactions, sleep, and family relationships being among the compromised factors [2,3,4]. One study estimated that almost three-quarters of children with chronic pain suffered physical impairment, half were absent from school, and more than a third had problems sleeping [5]. Moreover, pain frequency and intensity have also been shown to impact self-reported quality of life, with children and adolescents who experience extreme pain more often demonstrating impairments in psychological functioning, physical status, and functional status [6]. Additionally, as is the case with most pediatric chronic health conditions, the negative impact of chronic pain extends beyond the individual child affected, with parents and caregivers reporting that their own lives are greatly impacted and restricted [6,7].

### 1.1. The Unique Nature of Pediatric Chronic Pain

The WHO has emphasized the importance of understanding the physiological, developmental, and social uniqueness of pediatric chronic pain as distinct from adult chronic pain and the need for research examining pediatric populations specifically [1]. While it is known that chronic pain is common in children and adolescents, the prevalence rates vary widely in the literature [8]. It is known that certain sociodemographic characteristics play important roles in pediatric chronic pain. For example, a lower socioeconomic status is associated with a higher prevalence of pain in general, as well as a higher prevalence of headaches, which is the most common pain in children and adolescents [9]. Moreover, age and sex also appear to be associated with pediatric pain prevalence, with chronic pain being more common for females in their early teens [8,9,10].

### 1.2. Chronic Pain and Emotional, Developmental, or Behavioral Comorbidities

Pediatric chronic pain is often comorbid with psychological, developmental, and behavioral conditions, and has been linked to a greater risk of psychopathology both during childhood and adolescence, as well as into adulthood [11,12]. Moreover, even if chronic pain is resolved by adulthood, studies have shown that individuals who experienced pain in their youth were more likely to develop an anxiety or depressive disorder in their adult years [13]. Pediatric chronic pain has also been linked to behavioral problems, with the Biobehavioral Model of Pediatric Pain finding that a greater pain intensity and duration are associated with a greater risk of behavioral problems [14]. A 2016 narrative review on comorbidities in pediatric chronic pain highlighted the inextricable link between chronic pain and emotional, developmental, and behavioral comorbidities due to their shared neurobiology and mutually maintaining cognitive and behavioral factors [15].

### 1.3. Flourishing and Chronic Pain

While there is a large body of the literature on pediatric chronic pain and negative health outcomes, there are fewer works available in the literature on positive health indicators like flourishing [16]. There is no single definition of flourishing; rather, it is an umbrella term for the reduction in negative functioning and the increase in positive functioning including, but not limited to, optimism, resilience, psychological flexibility, acceptance, benefit finding, and hope [17]. Consistent with previous research, this study defined child flourishing as including curiosity, emotional regulation, and goal persistence [18]. These attributes were defined based on the work of an expert panel, supported by the Health Resources and Services Administration and facilitated by the Child and Adolescent Health Measurement Initiative in partnership with Child Trends. It was noted that these items were carefully chosen to ensure accuracy in parent reporting, to align with the established child flourishing models, to appropriately consider the developmental stages, and to minimize the survey’s burden on the respondents [18]. It is notable that the operational definition of flourishing does not address some important components of flourishing, including social relationships, which may result in different outcomes.

A number of studies have looked at similar constructs in children and adolescents with chronic pain and found a significant relationship between living with pain and resilience, social development, emotional adjustment, identity formation, and quality of life [19,20]. Notably, there are no known works in the literature on flourishing and children or adolescents with chronic pain plus an emotional, developmental, or behavioral comorbidity. Furthermore, the few studies that have been published have important limitations like small sample sizes and a lack of geographic diversity.

### 1.4. Objective and Hypotheses

Given the limitations of existing studies on pediatric chronic pain and positive health indicators, particularly regarding chronic pain and comorbidities, this study aims to use a large nationally representative sample to examine whether children and adolescents with chronic pain and children with chronic pain plus an emotional, developmental, or behavioral comorbidity demonstrate less flourishing than their typically developing peers. The developmental systems theory was used as the overall framework for this study, with development being conceptualized as a result of dynamic, bidirectional interactions between genes and environments across multiple systems [21]. The first research question was about the frequency in the health condition status groups among children aged 6–17 years in the United States. The second research question was about the sociodemographic characteristics among children aged 6–17 years across health condition status groups. The third research question was whether there was an association between the health condition status and flourishing in children aged 6–17 years. It was hypothesized that there would be a significant association such that as the chronic pain condition status is less severe, flourishing increases. The fourth research question was whether the health condition status would add significant incremental predictive utility in the prediction of flourishing over and above a block of demographic predictors. It was hypothesized that the health condition status would provide significant incremental predictive utility over and above a block of demographic predictors.

## 2. Methods

### 2.1. Data

Data were taken from the 2018–2019 National Survey of Children’s Health (NSCH), a nationally representative cross-sectional sample of children and adolescents conducted by the United States Census Bureau and funded by the United States Health Resources and Services Administration (HRSA). The NSCH randomly sampled households across the United States—via mailed surveys—who had at least one child between the age of 0 and 17 years. The overall response rate was 43.1% for 2018 and 42.4% for 2019. Data are publicly available online at no cost.

### 2.2. Participants

This study was limited to 6–17-year-olds (*N* = 31,435), with the sample being made up of 15,950 males (50.7%) and 15,485 females (49.3%). Sociodemographic factors of the selected sample were examined including the child’s sex, age, race/ethnicity, poverty level, whether the child comes from a working poor household, the highest level of parental education, and health insurance status.

### 2.3. Variables and Measures

#### 2.3.1. Health Condition Status Groups

Participants were grouped by health condition status. The first group, children experiencing chronic pain, was selected based on a parent reporting yes to “During the past 12 months, has this child had frequent or chronic difficulty with repeated or chronic physical pain, including headaches or other back or body pain?” and a parent reporting one special healthcare need (i.e., use of or need for prescription medication; above-average use of or need for medical, mental health, or educational services; and functional limitations compared with others of the same age that are not mental health related). The second group, children with chronic pain plus an emotional, developmental, or behavioral comorbidity (referred to hereafter as chronic pain plus), was selected based on the fulfillment of the above criteria for the chronic pain group and a parent reporting yes to “Does this child have any kind of emotional, developmental, or behavioral problem for which he or she needs treatment or counseling?” The third group, typical peers, was made up of participants aged between 6 and 17 years who did not meet the criteria for the chronic pain or chronic pain plus groups.

#### 2.3.2. Flourishing Categories

The outcome variable was the parent report of overall flourishing for children aged between 6 and 17 years, as well as responses to each of the three item-level flourishing criteria (i.e., shows interest and curiosity in learning new things, works to finish tasks they start, and can stay calm and in control when faced with a challenge). Individual items were coded as 1—“Always”; 2—“Usually”; 3—“Sometimes”; and 4—“Never.” Overall flourishing was coded with both three levels (1—“Always/usually response to 0–1 items”; 2—“Always/usually response to 2 items”; and 3—“Always/usually response to all 3 items”) and two levels (1—“Always/usually response to 0–2 items”; 2—“Always/usually response to all 3 items”). The two-level version of the variable was used for logistic regression.

#### 2.3.3. Predictors

Demographic variables used in the study included age (6–11 years/12–17 years), sex (male/female), and race/ethnicity (White/non-Hispanic, Hispanic, Black/non-Hispanic, other/multi-racial, or non-Hispanic). Family-based demographic variables included poverty level (0–99% Federal Poverty Level (FPL), 100–199% FPL, 200–399% FPL, and >400% FPL), highest level of parental education (less than high school, high school or General Educational Development Test (GED), some college or technical school, or college degree or higher), and current health insurance status (public insurance only, private health insurance only, public and private insurance, or uninsured). All variables were treated as categorical except age, poverty level, and parental education, which were treated as ordinal. Poverty level, a composite variable, was made up of responses to two questions, family income and number of family members.

### 2.4. Statistical Analysis

The analysis was conducted using the program IBM SPSS 27. To analyze research question 1, the frequencies of each health status condition in the sample were calculated to determine the prevalence of chronic pain plus, chronic pain, and typical peers. To analyze research question 2, for all health condition status groups, frequencies were calculated for each demographic variable included in the model (age, sex, race/ethnicity, poverty level, highest level of parental education, and current health insurance status). To analyze research question 3, tests of the association of demographic covariates and health condition status group were conducted using chi-squared test for categorical variables (sex, race/ethnicity, working poor status, and health insurance status) and Kendall’s tau-b for ordinal variables (age, parental education, and poverty level).

To analyze research question 4, the primary analysis, we conducted a hierarchical binary logistic regression with block model entry to determine whether health condition status (chronic pain plus, chronic pain, and typical peers) significantly predicted flourishing over and above demographic covariates. The dependent variable was two-level flourishing, with parent response of “Always/usually” to 0–2 items classified as “not flourishing”, and with “Always/usually” response to all 3 items classified as “flourishing”. The first block of predictors in the model included the variables sex, age, race/ethnicity, parental education, poverty level, and health insurance status. The second block of predictors added health condition status. This approach provided the opportunity to examine whether health condition status added to the explanation of parents’ perceived child flourishing (the outcome variable) over and above the contribution of the sociodemographic variables included in the model. For all analyses, we used a standard approach in social sciences for testing statistical probability at *p* < 0.05 as a way to identify statistically significant results.

## 3. Results

The estimated prevalence of chronic pain was 4.0%, and chronic pain plus was 3.9%, with typical peers making up 92.1%. The demographics for the sample and the results of the tests of association are shown in Table 1. There was a larger percentage of females than males in both the chronic pain and chronic pain plus groups. Across all three health condition groups, a larger percentage of participants were aged between 12 and 17 years. A majority of the sample across all three health condition groups was White/non-Hispanic. Tests of association between the health condition status and demographic variables were conducted. The hypotheses were that there would be significant associations such that a decreasing severity of the chronic pain condition status (i.e., chronic pain plus, chronic pain, and typical peers) would be associated with a younger age, being male, being White, having a higher socioeconomic status, a higher parental education, and a greater health insurance status. There were significant associations between the health condition status and all demographic variables in the expected directions.

Additional tests of association between the chronic pain condition status, using only the chronic pain plus and chronic pain categories of the variable, and the demographic variables were also conducted. This was potentially important because it could differentiate these two rarely studied groups. Again, the hypotheses were that there would be significant associations such that a decreasing severity of the chronic pain condition status (i.e., chronic pain plus and chronic pain) would be associated with being male, having a higher socioeconomic status, a higher parental education, and a greater health insurance status. The results were significant, in the hypothesized direction, and there were weak associations with sex, χ^2^ (2484) = 17.54, *p* < 0.001, Cramer’s *V* = 0.08; poverty level, τb (2484) = 0.06, *p* = 0.001; parental education; τb (2484) = 0.04, *p* = 0.04; and health insurance status, χ^2^ (2459) = 64.1605, *p* < 0.001, Cramer’s *V* = 0.16. But the results were not significant for age, race/ethnicity, and working poor. Of note, the chronic pain plus group was significantly less female (51.8%) than the pain group (60.1%).

Figure 1 shows flourishing across the three health condition status groups. There was a significant moderate association between the chronic pain condition status and flourishing (τb (31,405) = 0.249, *p* < 0.001), which indicates that this association is likely in the population. There was also a significant moderate association between the chronic pain condition status and the two-level version of flourishing (τb (31,405) = 0.237, *p* < 0.001).

Overall flourishing was made up of three items, and the association of the chronic pain condition status and each item of flourishing was significant in the expected direction as follows: shows interest and curiosity in learning new things, τb (31,354) = −0.145, *p* < 0.001, a weak association; works to finish the tasks they start, τb (30,936) = −0.177, *p* < 0.001, a weak association; and stays calm and in control when faced with a challenge, τb (31,196) = −0.207, *p* < 0.001, a moderate association. When examining only the chronic pain plus and chronic pain categories of the health condition status group variable, the association of the chronic pain condition status and each item of flourishing was also significant in the expected direction: shows interest and curiosity in learning new things, τb (2477) = −0.25, *p* < 0.001, a moderate association; works to finish the tasks they start, τb (2454) = −0.35, *p* < 0.001, a strong association; and stays calm and in control when faced with a challenge, τb (2468) = −0.41, *p* < 0.001, a strong association.

Hierarchical binary logistic regression was used to evaluate a model for predicting flourishing. The model was composed of two blocks of predictors. The first block of predictors consisted of age, sex, race/ethnicity, poverty level, parental education, and health insurance status. The chronic pain condition status constituted the second block of predictors.

The initial assumptions of logistic regression (independence of observations, categories of outcome variable, and categorical predictor variables are mutually exclusive and exhaustive, with a sufficient ratio of cases per predictor variable (here, >3000:10) were met. To assess multicollinearity among the predictors, all of which are categorical, the predictor variables were dummy coded, and a multiple linear regression was run with these dummy variables to generate VIF and tolerance values. The dummy variables were associated with some categories of the poverty level (tolerance = 0.23, VIF = 4.39), and the working poor families (tolerance = 0.28, VIF = 3.53) variables indicated that these two variables had a potentially problematic degree of multicollinearity. For all other predictor variables, the tolerance was >0.2 and the VIF was <1.6, indicating an absence of significant multicollinearity. After removing the working poor variable, the tolerance was >0.6 and the VIF was <1.7 for all predictor variables. Thus, the logistic regression analysis was repeated after excluding the working poor variable. As for the absence of outliers, leverage points, or highly influential points, 285 of 30,893 cases (0.92%) had a standardized residual value of > 2.5, with 4 of those cases having a standardized residual value of > 3.0. This was deemed to be a negligible portion of the sample; thus, the assumption was met.

The results of the hierarchical logistic regression are provided in Table 2. As expected, both the initial model, which consisted of the first block of predictors (age, sex, race/ethnicity, poverty level, parental education, and health insurance status), χ^2^ (15) = 1153.13, *p* < 0.001, and the full model, which consisted of the first and second block of predictors (chronic pain condition status), χ^2^ (16) = 2846.85, *p* = 0.000, significantly outperformed the null model. The Hosmer and Lemeshow test indicated that both the initial (χ^2^ (8) = 7.40, *p* = 0.49) and full models (χ^2^ (8) = 12.94, *p* = 0.11) fit the data appropriately. In addition, a pseudo R^2^ measure, used in logistic regression as a means to determine the goodness of fit of a model, yielded a Nagelkerke pseudo R^2^ = 0.054 for the initial model and a Nagelkerke pseudo R^2^ = 0.131 for the full model. This indicates that the full model predicted flourishing better than the initial model.

As expected, sex, age, race/ethnicity, poverty level, parental education, and health insurance status were all significant predictors of flourishing in the full model. Looking at the odds ratios for the pairwise comparisons for the demographic predictors in the full model, most were significant with a few exceptions. For the race/ethnicity variable, comparisons between White and Black (*p* = 0.757) and White and other/multi-racial, non-Hispanic (*p* = 0.145) races/ethnicities were not significant. For the health insurance status variable, the comparison between being uninsured and having both private and public health insurance (*p* = 0.280) was not significant.

As expected, the second block of predictors (chronic pain condition status) provided significant incremental utility in predicting flourishing (χ^2^ (2) = 1693.72, *p* < 0.001). In other words, adding the chronic pain condition status to the initial model increased the ability of the full model to predict flourishing over and above the other predictors. The classification statistics for both the initial and full models are provided in Table 3 and Table 4. The addition of the chronic pain condition status marginally increased the classification accuracy over the initial model from 75.4% to 77.8%.

Overall, the chronic pain condition status significantly predicted flourishing (Wald χ^2^ (1) = 1379.61, *p* < 0.001). Using the reciprocal of the odds ratios (and their confidence intervals) from Table 2, the odds of not flourishing for children with chronic pain is 2.33 times greater than for typical peers (OR = 2.33, 95% CI, 2.05 to 2.63). Of particular note, the odds of not flourishing for children with chronic pain and comorbidities is 13 times greater than for typical peers (OR = 12.99, 95% CI: 11.24 to 14.93).

## 4. Discussion

The results confirm the primary hypothesis that while chronic pain alone is problematic for flourishing, children and adolescents who also live with an emotional, developmental, or behavioral comorbidity have a much greater risk of not flourishing. These results are consistent with the findings of prior studies on both chronic pain and flourishing and chronic pain plus comorbidities and flourishing [15,18,19].

From a nationally representative dataset, chronic pain occurred in 4.0% of our sample, and the prevalence of chronic pain plus comorbidities was 3.9%. The overall chronic pain prevalence (7.9%) fell toward the lower end of the large prevalence range (6–57%) identified in the existing literature base [8,22]. However, in this study, we identified children and adolescents with chronic pain through parent endorsement of the criterion “this child had frequent or chronic difficulty with repeated or chronic physical pain, including headaches or other back or body pain” over the past 12 months, plus parent report of at least one special health need (e.g., use of or need for prescription medication; above average use of or need for medical, mental health, or educational services; functional limitations compared with others of the same age that are not mental health related). This allowed us to identify children who were functionally limited by their chronic pain and/or required the help of professional services. To identify those with comorbidities, parent response to the question “Does this child have any kind of emotional, developmental, or behavioral problem for which he or she needs treatment or counseling?” was used in addition to the chronic pain criteria outlined above. This was a different approach to identifying children and adolescents with chronic pain who were functionally limited by their chronic pain, with comparable studies identifying participants with chronic pain exclusively using parental report of chronic pain in the last 12 months. Therefore, our more nuanced and stringent inclusion criteria may be the reason why our overall prevalence rate was lower than that reported in other studies [22].

Identifying and managing chronic pain and associated comorbidities is extremely important in both the short and long term. In addition to pain-related functional impairment, poorer social, emotional, physical, and socioeconomic outcomes are more likely to be experienced by those who experience chronic pain during childhood than typical peers [2,3,13]. These negative outcomes not only burden the individuals, but also their families, immediate communities, and the greater population. Therefore, healthcare systems must work to reduce chronic pain and the commonly associated emotional, developmental, or behavioral comorbidities. Moreover, given the potentially more deleterious outcomes for those with chronic pain and comorbidities as identified by this study and other recent similar studies, resources are needed to evaluate and treat chronic pain effectively in pediatric populations [23]. The even greater risk of not flourishing for those in the chronic pain and comorbidities group identified by the results of this study underscores the need for more consistent screening that considers mental health, as well as resiliency factors, in pediatric patients. All too often, screenings are focused exclusively on medical factors or current pain status and fail to consider other needs.

The developmental systems theory underpinned this study, as both chronic pain and flourishing are complex and multidimensional, with several factors playing important roles in their relationship [21]. Looking at the sociodemographic variables included in the model (age, sex, race/ethnicity, poverty level, parental education, and health insurance status), there was a significant association between the chronic pain condition status and demographic variables. Looking only at the chronic pain plus and chronic pain groups, the results were significant for sex, poverty level, parental education, and health insurance status but not for age, race/ethnicity, and working poor families. While many of these demographic characteristics are consistent with previous research related to risk and resiliency, it is important for practitioners to uniquely consider parental education and gender [24]. Notably, as seen in the literature, those in the chronic pain group were more likely to be female (60.1%) than male (39.9%); however, the chronic pain plus group did not have such a substantial sex disparity, with males and females each representing close to 50%. This is an important finding for practitioners, as male children tend to be overlooked relative to emotional screening [25].

### 4.1. Implications

The results of this study highlight a clear need for practitioners to improve the universal screening of pediatric chronic pain and common mental health comorbidities. By accurately identifying the children and adolescents who are impacted by chronic pain, interventions can be focused not only on reducing negative outcomes, but also on promoting positive outcomes like flourishing. Moreover, when assessing pediatric pain, those with mental health comorbidities must be identified, as traditional chronic pain treatments may not be sufficient to help facilitate positive outcomes for this population. Healthcare professionals need to be educated regarding the fact that chronic pain treatments may not be as successful if they do not consider mental health comorbidities. The uniqueness of those with chronic pain plus an emotional, developmental, or behavioral comorbidity highlighted in the results also underscores the need for the development of specific interventions that target chronic pain as well as mental health comorbidities in pediatric populations.

The results also emphasize the need for more evidence-based interventions specifically targeted at low-income low-education families. Given the negative outcomes that can occur later in life if chronic pain is not addressed during childhood, it is particularly important that healthcare professionals who work with the populations who are most impacted, as well as parents, caregivers, and teachers in these communities, are educated about how to identify and treat chronic pain [4]. Education programs must be mindful of those children whose parents have a lower level of education though, and they must consider how to make psychoeducation accessible and subsequently impactful to all children living with chronic pain.

### 4.2. Limitations

The findings of this study should be assessed with the following limitations in mind. Firstly, chronic pain was identified exclusively with the parent responses to two questions based exclusively on the child’s health and behavior over the last 12 months. While this allowed us to identify children and adolescents who were experiencing chronic pain that was impacting their functioning or requiring medical attention, this was a more specific criteria than what has been used by comparable studies, which resulted in a smaller reported prevalence. Moreover, given the cross-sectional study design, it was not possible to infer a causal pathway between chronic pain condition status and flourishing. This relationship may be bidirectional, with a lack of flourishing leading to chronic pain and emotional, developmental, and behavioral comorbidities.

## 5. Conclusions

Ultimately, our findings suggest that while children and adolescents living with chronic pain are less likely to flourish than typical peers, those who live with chronic pain plus an emotional, developmental, or behavioral comorbidity are even more vulnerable to reduced flourishing. Therefore, there is an urgent need to provide specialized care to these children and to expand the scope of chronic pain screening and treatment to include common mental health comorbidities.

## Figures and Tables

**Figure 1 children-10-01531-f001:**
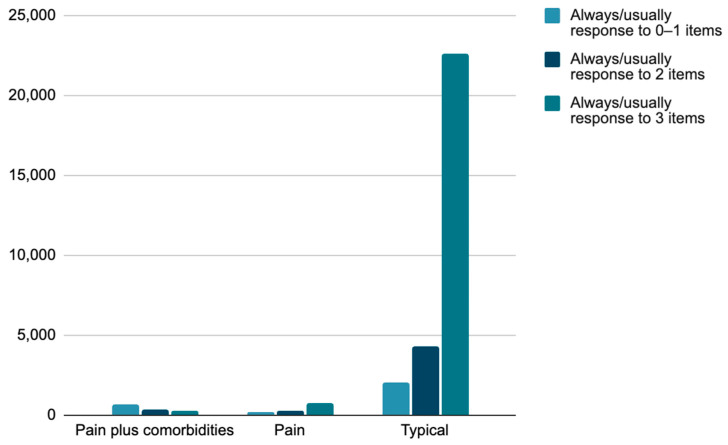
Flourishing in chronic pain plus, chronic pain, and typical groups.

**Table 1 children-10-01531-t001:** Demographic table.

Characteristic	Chronic Pain Plus	Chronic Pain	Typical Peers	χ^2^	Cramer’s *V*
Age, years				−0.10 ^abc^	0.10
6–11 years	355 (27.9%) ^c^	315 (26.0%)	12,991 (44.9%)		
12–17 years	918 (72.1%)	896 (74.0%)	15,960 (55.1%)		
Sex				63.98 ^b^	0.05
Male	614 (48.2%) ^c^	483 (39.9%)	14,853 (51.3%)		
Female	659 (51.8%)	728 (60.1%)	14,098 (48.7%)		
Race/Ethnicity				43.98 ^b^	0.03
White, non-Hispanic	907 (71.2%)	872 (72.0%)	19,912 (68.8%)		
Hispanic	143 (11.2%)	121 (10.0%)	3485 (12.0%)		
Black non-Hispanic	110 (8.6%)	97 (8.0%)	1832 (6.3%)		
Other/multi-racial, non-Hispanic	113 (8.9%)	121 (10.0%)	3722 (12.9%)		
Working Poor				9.02 ^d^	0.02
Working Poor	111 (8.9%)	118 (9.9%)	2192 (7.7%)		
Not Working Poor	1140 (91.1%)	1075 (90.1%)	26,097 (92.3%)		
Poverty Level				0.08 ^ab^	0.07
0–99% FPL	248 (19.5%)	198 (16.4%)	3013 (10.4%)		
100–199% FPL	304 (23.9%)	246 (20.3%)	4456 (15.4%)		
200–399% FPL	357 (28.0%)	369 (30.5%)	9043 (31.2%)		
>400% FPL	364 (28.6%)	398 (32.9%)	12,439 (43.0%)		
Parental education				0.06 ^ab^	0.05
Less than high school	41 (3.2%)	36 (3.0%)	823 (2.8%)		
High school or GED	206 (16.2%)	204 (16.8%)	3846 (13.3%)		
Some college or technical school	434 (34.1%)	340 (28.1%)	6704 (23.2%)		
College degree or higher	592 (46.5%)	631 (52.1%)	17,578 (60.7%)		
Health Insurance				887.64 ^b^	0.12
Public insurance only	466 (36.9%)	328 (27.4%)	4727 (16.6%)		
Private health insurance only	606 (48.0%)	750 (62.7%)	21,454 (75.4%)		
Public and private insurance	152 (12.0%)	75 (6.3%)	742 (2.6%)		
Uninsured	38 (3.0%)	44 (3.7%)	1536 (5.4%)		

^a^ Kendall’s tau-b used, ^b^
*p* < 0.001, ^c^ percentage of sample, ^d^
*p* < 0.01.

**Table 2 children-10-01531-t002:** Results of logistic regression (full model).

Variable	*b*	*SE*	Wald	*p*	OR	95% CI	
						Lower	Upper
Constant	−0.784	0.096	66.67	<0.001	0.457		
Sex (Male) ^a^	−0.308	0.028	119.81	<0.001	0.736	0.696	0.777
Age (6–11 years)	−0.250	0.028	77.73	<0.001	0.779	0.737	0.823
Race/ethnicity (White)			20.85	<0.001			
Hispanic	−0.173	0.043	16.01	<0.001	0.841	0.773	0.916
Black	−0.018	0.057	−0.10	0.757	0.983	0.879	1.098
Other/multi-racial, non-Hispanic	0.064	0.044	2.12	0.145	1.066	0.978	1.162
Poverty (400% FPL or greater)			50.38	<0.001			
0–99% FPL	−0.267	0.055	23.87	<0.001	0.766	0.688	0.852
100–199% FPL	−0.228	0.047	24.07	<0.001	0.796	0.727	0.872
200–399% FPL	−0.230	0.035	42.99	<0.001	0.794	0.742	0.851
Parent education (college degree or higher)			172.18	<0.001			
Less than high school	−0.843	0.080	111.69	<0.001	0.430	0.368	0.503
High school or GED	−0.404	0.044	84.53	<0.001	0.668	0.612	0.728
Some college or technical school	−0.293	0.035	69.08	<0.001	0.746	0.696	0.799
Health insurance (uninsured)			63.76	<0.001			
Public health insurance only	−0.132	0.065	4.16	0.041	0.876	0.771	0.995
Private health insurance only	0.192	0.062	9.69	0.002	1.212	1.074	1.368
Public and private insurance	−0.100	0.093	1.17	0.280	0.904	0.754	1.085
Pain condition status (typical)			1379.61	<0.001			
Chronic pain plus	−2.560	0.072	1259.52	<0.001	0.077	0.067	0.089
Chronic pain	−8.44	0.063	178.68	<0.001	0.430	0.380	0.487

^a^ Reference category.

**Table 3 children-10-01531-t003:** Summary of classification statistics from hierarchical logistic regression analysis.

Variable	Initial Model	Full Model
Accuracy	75.4%	77.8%

**Table 4 children-10-01531-t004:** Classification statistics from hierarchical logistic regression analysis (full model).

	Not Flourishing	Flourishing	Percentage Correct
Observed			
Not flourishing	1156	6451	15.2%
Flourishing	401	22,885	98.3%
Overall percentage			77.8%

## Data Availability

The National Survey of Children’s Health is a publicly available dataset funded by the Health Resources and Services Administration and the Maternal and Child Health Bureau. Data are publicly available at https://www.childhealthdata.org/ (accessed on 1 March 2023).

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
