# Peer review of "Flourishing among Children and Adolescents with Chronic Pain and Emotional, Developmental, or Behavioral Comorbidities"

_children, 2023, doi:10.3390/children10091531_

Round 1
Reviewer 1 Report
Flourishing Among Children and Adolescents with Chronic Pain and Emotional, Developmental, or Behavioral Comorbidities
Thank you very much for allowing me to review this article. It addresses a topic of interest, however there are certain aspects that must be considered.
Abstract:
• Not enough information is provided on the analysis of the data and on the findings found.
Introduction:
• I think the introduction is too long. It could be summarized and include only the most relevant information.
methods:
• Indicate here what the response rate was
• How was the randomization done for the sending of the emails?
• Specify and provide information on the composite variables that have been taken into account, such as FPL
• The statistical analysis is not adequately specified.
Author Response
Dear reviewer,
We sincerely appreciate your thoughtful comments and feedback. We have answered each of your bulleted points below:
Abstract:
- Not enough information is provided on the analysis of the data and on the findings found. – We very much appreciate this feedback. We have edited the abstract to include more information on the analysis and findings.
Introduction:
- I think the introduction is too long. It could be summarized and include only the most relevant information. – Thank you very much for this suggestion, we have edited the introduction to make it more succinct.
Methods:
- Indicate here what the response rate was. – Thank you for raising this important point, we have included this information in the methods section.
- How was the randomization done for the sending of the emails? – Thank you for bringing out attention to the fact that this was not clear in the manuscript. The surveys were mailed to participants. We have clarified this in the manuscript.
- Specify and provide information on the composite variables that have been taken into account, such as FPL – Thank you very much for this suggestion, we have clarified this in the methods section of the manuscript.
- The statistical analysis is not adequately specified. – We are very open to feedback on the statistical analysis performed. Are you able to please be more specific as to what you are looking for? We are not sure what this pertains to exactly.
Thank you again for taking the time to review our work. We really appreciate your help in enhancing our manuscript.
Reviewer 2 Report
Thank you for the opportunity to review this interesting study examining the relationship between paediatric chronic pain and the psychological construct of flourishing. The study draws on data collected in the 2018/2019 National Survey of Child Health in the USA and uses this data to examine the relationships between certain demographic variables and flourishing in children aged 6 – 17 years with ‘repeated or chronic physical pain.’ A strength of this paper is its clearly reported and large random sample, definitions used, and the analytic strategy. The authors establish the rationale for this study based on the impact of chronic pain on the developmental and future functioning of children and adolescents, while also pointing out the impact on family and caregivers. This area of research is particularly important because of the long-term impact difficulties during childhood and adolescence have on adult participation.
The study uses data collected for a different purpose, and therefore does not have a stand-alone measure of flourishing. Flourishing is an umbrella term for positive attributes of health with Seligman (2011) promoting the “PERMA” model (Positive emotions, Engagement, Relationships, Meaning, Achievements) as the constituents of flourishing. Its measurement is hotly debated, as is the construct itself, but it has been widely recognised as having a significant impact on quality of life and participation. The items the authors selected were drawn from the National Survey of Children’s Health and included four: an overall report of flourishing (as reported by parents), and three items ‘shows interest and curiosity in learning new things’, ‘works to finish tasks they start’, and ‘can stay calm and in control when faced with a challenge.’ I note that an earlier study using data from the 2011/2012 NSCH survey used the same items (Kandasamy et al., 2018).
It would be helpful to include some further discussion about flourishing and its measurement to help justify the items selected and embed the discussion within the larger discourse about flourishing in children and adolescents. While the Kandasamy et al (2018) paper briefly indicates that these three items were developed “… based on a review by a technical expert panel, which included experts in the fields of survey methodology, children's health, community organizations and family leaders, and incorporated public comment” this is not mentioned in the current paper. If the PERMA model is considered, then relationships and meaning don’t appear to be measured (eg there doesn’t appear to be any measure of social participation, friendships etc). As I am not an expert in paediatric pain and flourishing, I cannot comment on the adequacy of the items selected, but I would have expected that social interaction/engagement would be an important component of flourishing, as it is in adults.
Strengths
This study draws on a very large cohort of participants, with representative random sampling across the United States of America, and clearly identifies that socio-economic and ethnicity are associated with poorer health status, flourishing and increased chronic pain. Stratifying the chronic pain groups into ‘chronic pain’ and ‘chronic pain plus comorbidities’ is insightful, and points to a large group of young people with complex needs that according to this study may need specialised support. I was especially pleased to note the authors highlighting the need for male children to be screened for emotional problems as the sex disparity between males and females was not significant for children with both chronic pain and comorbidities. It is concerning that sex disparities exist at all ages with females over-represented.
From my understanding of the analytic strategies employed, these were appropriate. I am not a statistician, and I would recommend the analysis be reviewed by a biostatistician.
I found the manuscript well-written, comprehensive (excepting the definition of flourishing), and well-structured. The paper is highly relevant for understanding chronic pain and child development/flourishing.
I found the references useful, accurate and up-to-date.
Kandasamy, V. , Hirai, A. , Ghandour, R. & Kogan, M. (2018). Parental Perception of Flourishing in School-Aged Children: 2011–2012 National Survey of Children's Health. Journal of Developmental & Behavioral Pediatrics, 39 (6), 497-507. doi: 10.1097/DBP.0000000000000559.
M.2011New York, NYFree Press
Author Response
Dear reviewer,
We sincerely appreciate your thoughtful and detailed feedback. We specifically thank you for the important emphasis you placed on ensuring that flourishing is adequately discussed in the introduction section of our paper.
The codebook for the National Survey of Children’s Health 2018-2019 defines the variable ‘flourishing’ as a composite variable that aims to “capture curiosity and discovery about learning, resilience, and self-regulation.” As you mentioned, the variable was defined following a “review of positive health indicators by a Technical Expert Panel.” In addition, there was a “public comment period which yielded more interest in this concept.” We appreciate you pointing out that this important information is missing from our manuscript. We have added more detail to clarify how flourishing was defined and measured in the study.
The PERMA model was not used for this paper as the data we used already set forth a defined model for measuring flourishing. We thank you for putting forward this important point though.
Thank you again for your time and expertise.
Reviewer 3 Report
Dear authors,
thank you for the opportunity to have an insight into your manuscript Flourishing Among Children and Adolescents with Chronic Pain and Emotional, Developmental, Behavioral Comorbidities. This is a very interesting manuscript and I believe it can be further improved.

Author Response
Dear reviewer,
We sincerely appreciate your thoughtful comments and feedback. We have answered each of your bulleted points below:
- The introductory part of the manuscript is presented very nicely and in detail, however, I wonder if all the important information can be contained in one paragraph, since such a presentation is more common. In this way, the introduction seems rather long compared to other parts of the manuscript, the discussion for example. – Thank you very much for this suggestion, we have edited the introduction to make it more succinct.
- The part 1.4. Objective and Aims would rather call it objectives and hypotheses and I also think it can be the final part of the Introduction. I propose to eliminate a part of the text that is repeated and was already mentioned earlier. – Thank you for this suggestion, we agree. We have renamed the section and edited the information to not repeat information that it mentioned elsewhere in the paper.
- Line 166 – GED - need to clarify. I assume it's an abbreviation. – Thank you for pointing this out, we have added the full term.
- Line 188-189 - A majority of the sample across all three pain status groups were White, non-Hispanic - is it true that there are three groups of pain status? Typical Peers do not seem to me to belong to status pain. – Thank you very much for bringing this to our attention. We agree that this could be made more clear in the manuscript. There were three groups, typical peers, chronic pain, and chronic pain plus. We have now changed the terminology in the manuscript to “Health Condition Status” rather than “Chronic Pain Status” to clarify this.
- The part of the text that is now in the Results and refers to the research questions, rather state in the previous chapter Methods and in the chapter of results only the concrete results of the conducted analyses. Although this kind of presentation may make the results clearer in the field of biomedicine, it is not recommended or common. – Thank you very much for this suggestion, we have actioned this feedback and included only the concrete results of the analyses in the results section.
- In Table 1 the results shown in age and sex are identical. Is it a mistake maybe? Please check. – Thank you very much for pointing this out, we have updated the results shown in the table.
- Also please double check the percentages. In terms of percentages, it seems to me that in some places the entire examined sample is not covered. – Again, thank you very much for pointing this out, we have updated the results shown in the table.
- In the chapter Discussion line 300-301 it is not usual to write we found.... Rather state it was found……. – Thank you for this suggestion, this has been amended throughout the manuscript.
- I also suggest that you do not repeat the specific results figures that are already listed in the results chapter, but that you interpret them in the Discussions chapter. – Thank you for this important suggestion, we have edited the discussion accordingly.
- If possible, replace references older than 5 years with more recent references. – Thank you very much for this important suggestion, we have gone through the references and updated as many as possible to be more current. A majority of the references are from the last 5 years, however, to adequately reflect the literature available on this topic, some pivotal references from more than 5 years ago have been kept in the manuscript.
Again, thank you very much for taking the time to provide such comprehensive and important feedback. We really appreciate your help in enhancing our manuscript.
Round 2
Reviewer 1 Report
Thank you very much for allowing me to review the article again. I think it has improved substantially. However, let me insist that the data analysis section is not adequately detailed. Please, consult other articles of similar methodology so that they have it as a model. It should be better specified how the regression model was built, among other aspects.
Author Response
Dear reviewer,
Thank you again for your thoughtful and important feedback. We consulted a number of articles that utilized a similar methodology and elaborated on our statistical analysis section per your recommendation.
Again, we sincerely appreciate your help enhancing our manuscript.